# Microstructural, Tribology and Corrosion Properties of Optimized Fe_3_O_4_-SiC Reinforced Aluminum Matrix Hybrid Nano Filler Composite Fabricated through Powder Metallurgy Method

**DOI:** 10.3390/ma13184090

**Published:** 2020-09-15

**Authors:** Negin Ashrafi, M. A. Azmah Hanim, Masoud Sarraf, S. Sulaiman, Tang Sai Hong

**Affiliations:** 1Department of Mechanical and Manufacturing Engineering, Universiti Putra Malaysia, Serdang 43400, Malaysia; ashrafinegin2000@gmail.com (N.A.); shamsuddin@upm.edu.my (S.S.); saihong@upm.edu.my (T.S.H.); 2Research Center Advance Engineering Materials and Composites (AEMC), Faculty of Engineering, Universiti Putra Malaysia, Serdang 43400, Malaysia; 3Centre of Advanced Materials, Department of Mechanical Engineering, Faculty of Engineering, University of Malaya, Kuala Lumpur 50603, Malaysia; masoudsarraf@gmail.com; 4Materials Science and Engineering Department, Sharif University of Technology, P.O. Box 11155-9466, Azadi Avenue, Tehran, Iran

**Keywords:** aluminum matrix composite, hybrid composite, corrosion rate, tribology, Fe_3_O_4_-SiC

## Abstract

Hybrid reinforcement’s novel composite (Al-Fe_3_O_4_-SiC) via powder metallurgy method was successfully fabricated. In this study, the aim was to define the influence of SiC-Fe_3_O_4_ nanoparticles on microstructure, mechanical, tribology, and corrosion properties of the composite. Various researchers confirmed that aluminum matrix composite (AMC) is an excellent multifunctional lightweight material with remarkable properties. However, to improve the wear resistance in high-performance tribological application, hardening and developing corrosion resistance was needed; thus, an optimized hybrid reinforcement of particulates (SiC-Fe_3_O_4_) into an aluminum matrix was explored. Based on obtained results, the density and hardness were 2.69 g/cm^3^, 91 HV for Al-30Fe_3_O_4_-20SiC, after the sintering process. Coefficient of friction (COF) was decreased after adding Fe_3_O_4_ and SiC hybrid composite in tribology behaviors, and the lowest COF was 0.412 for Al-30Fe_3_O_4_-20SiC. The corrosion protection efficiency increased from 88.07%, 90.91%, and 99.83% for Al-30Fe_3_O_4_, Al-15Fe_3_O_4_-30SiC, and Al-30Fe_3_O_4_-20SiC samples, respectively. Hence, the addition of this reinforcement (Al-Fe_3_O_4_-SiC) to the composite shows a positive outcome toward corrosion resistance (lower corrosion rate), in order to increase the durability and life span of material during operation. The accomplished results indicated that, by increasing the weight percentage of SiC-Fe_3_O_4_, it had improved the mechanical properties, tribology, and corrosion resistance in aluminum matrix. After comparing all samples, we then selected Al-30Fe_3_O_4_-20SiC as an optimized composite.

## 1. Introduction

In the past few decades, focus on materials to enhance the comprehensive performance of aircraft, automotive, and marine component parts has motivated the industrial sectors to improve on composite materials [1,2]. Among the various kinds of composite, hybrid aluminum metal matrix composites are being widely selected to fulfil the industrial requirements [3,4]. Hybrid aluminum matrix composites (HAMC) are the next-generation composites that can replace single reinforcement composites and introduce new features to improve the performance of these materials [5]. According to previous researches, adding nanoparticles to the composite affects its properties [6,7]. The wear resistance and strength of the hybrid reinforcements are much higher than for aluminum [8]. Properties of the composite mostly depend on reinforcement weight percentage, chemical reaction with matrix, the grain size of reinforcement, and the production method [9]. Hybrid aluminum metal matrix composites can be fabricated through infiltration, powder metallurgy, squeeze casting, semi-solid, and stir casting [10].

In a broad variety of industrial applications, hybrid aluminum matrix composites can be considered as replacement of conventional material due to attractive properties, including excellent corrosion resistance, high strength-to-weight ratio, lower thermal expansion coefficient, good casting ability, lower density, higher strength, greater wear resistance, better fatigue resistance, and improved stability at elevated temperature [11,12,13]. Several researches have been carried out on composites with two or more reinforcements, such as SiC particles with carbon nanotubes (CNT) or Al_2_O_3_ reinforced aluminum matrix composites (AMC), to fabricate hybrid aluminum matrix composites focusing on the investigation of hardness, strength, wear, and thermal properties [14].

Previous research confirmed that AMC is a suitable composite for an extensive diversity of applications, since it is an excellent multifunctional lightweight material with remarkable properties; however, aluminum alloys are required to improve their wear resistance in high-performance tribological application because of their poor wear resistance and low hardness value [15]. Therefore, there is a high possibility of oscillation if these materials were joined by bolting or riveting, which might lead to sliding wear in some environmental conditions. Thus, reinforcement of particulates can be used to overcome this problem; these reinforcements, including silicon carbide, are famous for their high-specific strength, which leads to the use of metal matrix composites to address the demand for wear and corrosion resistance material [16]. There is also a demand for highly wear-resistive materials. Investigation in this area is vital from the economic perspective as cost is directly involved [17].

Moreover, appropriate material selection is crucial in producing superior product with high cost-efficiency. Silicon carbide (SiC), being a high-powered semiconductor with wide band gap and high electron-mobility, is broadly selected as reinforcing particles in aluminum composite. Silicon carbide particles have great mechanical properties, high thermal conductivity, low coefficient of thermal expansion (CTE), and low market price [18]. Magnetite (Fe_3_O_4_) is also a suitable filler due to its low-cost and higher free energy thermite reaction with aluminum. This reaction can improve the wettability between magnetite and aluminum matrix in offering extra energy for the process, and it is a paramount characterized filler material due to its excellent magnetic properties [19].

Furthermore, corrosion behavior is an essential indicator for consideration in the application of composites as structural materials. Reinforcing particulates might interact chemically, physically, or electrochemically with the matrix and accelerate the corrosion rate [20]. Moreover, galvanic interactions among the matrix and reinforcement can also increase the corrosion rate. Several corrosion investigations were carried out of Al matrix composites, pertaining on the corrosion susceptibility in NaCl. Corrosion-resistance improvement of the AMC was reported due to the increase in the SiC volume and optimized amount of SiC by different research work. Boutouta et al. highlighted that the Taffel extrapolations indicate a singularity, as shown by the sample containing 40% Fe_2_O_3_ which had the best electrochemical performance due to its lowest corrosion rate and the lowest I*corr*. However, there is no research focusing on the corrosion properties of Al-Fe_3_O_4_-SiC [21].

The objective of this research is to fabricate the HAMC with addition of different weight percentage in Fe_3_O_4_ and SiC nanoparticles as reinforcement, to find the optimum amount of Fe_3_O_4_ and SiC addition into this hybrid composite. Microstructure, hardness, tribological, and corrosion properties of the composite were also assessed. It is worth mentioning that the knowledge obtained in this research will contribute to the development of a novel hybrid composite with relation to finding optimum amount of nanoparticles filler which can be considered for various applications.

## 2. Materials Selection and Method

### 2.1. Fabrication Process

Pure aluminum powder with a purity of 99.7% and an average particle size of 20 μm was used as the composite matrix. Commercially available SiC with an average grain size of 2 μm and Fe_3_O_4_ (45–70 nm) purchased from (MHC Industrial Co., Ltd., China) was used to reinforce the Al matrix. The fabrication procedures were diligently carried out by firstly mixing the SiC and Fe_3_O_4_ powders, and Al matrix particles were mechanically milled for 2 h, using a planetary ball mill (PM 100, Retsch, Haan, Germany), at a speed of 400 rpm, at room temperature. The milling-ball-to-powder weight ratio was at 15:1. Proper mixing is essential in the powder metallurgy process. The blended powder with a binder (Mg Stearate) was discharged into a tubular die (diameter of 20 mm). During ball-milling, adding magnesium stearate can avoid agglomeration of particles and improve the distribution of reinforcements in the structure. Secondly, using a universal testing machine (Instron 3382) compaction attains green compacts of powder, and then cold-iso-pressed (CIP) in one direction at a pressure of 2500 Kgf/cm for 15 min, to attain an initial green density ranging from 85 to 95%. By using a Linn High Therm furnace, we heated the compacts at 600 °C. The sintering process was done under argon atmosphere, to prevent oxidation, and the temperature was fixed at 600 °C for 20 min, with heating and cooling rate of 5 °C/min, and then soaked in the furnace for 24 h. Eight basic composition mixtures of magnetic nano iron oxide and silicon carbide were as presented in Table 1. All of the compositions comprise 5% Mg Stearate powder. For evaluation purposes of the microstructural characterization, we prepared specimens by grinding on various abrasive papers of 800, 1200, 2000, and 2500 grit and polishing with diamond paste, using alumina slur and ultrasonic cleaning in acetone and deionized water, for 10 min, and drying at 100 °C for 1 h.

### 2.2. Characterization

#### 2.2.1. Phase Analysis and Microstructural Characterization

A PANalytical Empyrean system (Grazing incidence X-ray diffraction (GIXRD), The Netherlands) was utilized for phase analysis with Cu–Kα radiation (λ = 1.54178 Å), over a 2θ range from 20° to 80°, operating at 45 kV and 30 mA, with a scanning rate of 0.1°s^−1^ and step size of 0.026°. To investigate the XRD patterns, the “PANalytical X’Pert HighScore” software was also deployed to wherein all the reflections assimilated with the standards collected by the Joint Committee on Powder Diffraction and Standards (JCPDS, card 02-1109 for Al, 01-075-1609 for Fe_3_O_4_, and 019-1138 for SiC). Field-emission scanning electron microscopy (FESEM, SU8000, Hitachi, Japan), with an acceleration voltage of 2 kV, was used to reveal the particle morphologies and microstructure. The microstructural investigation focused on the surface of specimens. Energy-dispersive spectroscopy (EDS) attached to the FESEM machine was used to perform elemental analysis.

#### 2.2.2. Micro Hardness

The device used for evaluating the micro-hardness was a Vickers micro hardness testing Machine (Mitutoyo-AVK C200-Akashi Corporation, Kanagawa, Japan) with the total load selected at 98.07 mN and a dwell time of 15 s. The tests were conducted on 5 diverse random areas, and the mean value was then calculated for each sample composition.

#### 2.2.3. Wear Test

By utilizing a pin-on-disc configuration (Ducom Reciprocating Friction Monitor-TR 282 Series), we performed the wear tests in dry-sliding condition. This machine is used to measure wear characteristics and the friction of the specimens, through reciprocating sliding movement. A reciprocating engine is utilized to generate a bi-directional sliding movement between the samples, while a loading mechanism applies the chosen load upon the test samples. Moreover, the instant friction force by a friction measurement system can be measured. Coefficient of friction (COF) and a diversity of optional facilities are also measured and demonstrated on the “WinDucom” software. The dry-sliding experiment starts as the alumina cylindrical pin, in 6 mm diameter and 8 mm length, glides against a stationary counterpart plate. Before the wear test, both pins and samples were cleaned with distilled water and degreased with acetone. The normal loads of 10 N are kept constant, while a reciprocating frequency of 10 Hz and amplitude stroke of 1 ± 0.02 mm were applied to the disc, where the tangential frictional force was continuously calculated by using a load cell sensor attached to the pin-holder arm and recorded in a root mean square value. The kinetic coefficient of friction (*μ*_k_) of each sample during 150 s duration was produced in the instrumentation output, which was determined by dividing the recorded frictional force by the normal load. Besides this, an atomic force microscope (AFM, Ambios Technology) was used to evaluate the topographical texture of the surfaces and wear scars (tribo-path).

#### 2.2.4. Corrosion Behavior

The standard three-electrode configuration designated as working electrode, counter electrode, and reference electrode was utilized in potentiodynamic polarization measurements. In this study, the reference electrode used was a saturated calomel electrode (SCE), composite samples were used as the working electrode, and graphite as counter electrode.

The electrolyte used during the entire experiment was artificial seawater prepared at room temperature. Based on the Burkhoder’s formulation B, the compositions of the simulated seawater were as follows (per liter): 23.476 g NaCl + 3.917 g Na_2_SO_4_ + 0.192 g NaHCO_3_ + 0.664 g KCl + 0.096 g KBr + 10.61 g MgCl2·6H2O + 1.469 g CaCl_2_·6H_2_O + 0.026 g H_3_BO_3_ + 0.04 g SrCl_2_·6H_2_O + 0.41 g MgSO_4_·7H_2_O + 0.1 g NH_4_Cl + 0.1 g CaSO_4_ + 0.05 g K_2_HPO_4_ + 0.5 g tri-sodium citrate + 3.5 g sodium lactate + 1 g yeast extract. The pH was then adjusted to 7.5 ± 0.1, using a 5 M NaOH solution [22].

The surface zone exposed into the electrolyte was 1 cm^2^. To collect and assess the experiment data, a potentiostat Bio-Logic SP-150 tracked by a PC computer and EC-Lab software were utilized. The potential ranges from −2000 to +2000 mV versus SCE reference electrode were plotted in the potentiodynamic polarization curves. The scanning rate through the experiment was 1 mVs^−1^. A duration of 30 s was applied at the beginning of the experiment to obtain the steady-state testing condition. The tests were repeated at the same conditions for three times to confirm the consistency of the data obtained.

Tafel plots were then obtained, thus enabling us to extract essential information, such as corrosion potentials (*E_corr_*/V_SCE_) and corrosion current (*I_corr_*/μA cm^−2^). By using the following equation [23], the corrosion protection efficiency (*P.E.*) can be calculated:(1)$$P.E.(\%)=\frac{I_{corr}^{0}-I_{corr}^{c}}{I_{corr}^{0}}\times 100$$
where $I_{corr}^{0}$ is the corrosion current of the Al-15Fe_3_O_4_, and $I_{corr}^{c}$ is the corrosion current after adding different wt% reinforcement of Fe_3_O_4_ and SiC to the composite.

Moreover, the *CR* was measured by using the following formula [24]:(2)$$CR({\mathrm{mm}\;\mathrm{year}}^{-1})=\frac{0.13I_{corr}(E.W.)}{d}$$
where *d*, *E.W.*, and *I*_corr_ are density of the corroding species in g cm^−3^, weight of the corroding species in g, and the corrosion current in A cm^−2^, respectively.

## 3. Results and Discussion

### 3.1. Microstructural Evaluation

The optical microscopy images for Al-15Fe_3_O_4_ and Al-30Fe_3_O_4_-20SiC composites are presented in Figure 1a,b, respectively. Figure 1a shows the homogenous distribution of Fe_3_O_4_ particles in the Al matrix. Figure 1b shows SiC as a gray-color element, and Fe_3_O_4_ particles are the white-color element which is distributed quite uniformly in the Al-30Fe_3_O_4_-20SiC sample.

Efficient reinforcement requires a well-bonded principal with matrix and particles. Chemical bonding (covalent, metallic, and ionic) inter-diffusion is the diffusion of atoms between two metals, and van der Waals bonding refers to the components of the interface mechanisms pertaining to the filler and matrix bonding, and the reaction among the matrix and reinforcements in composite.

The appropriate reaction among matrix and reinforcements assists wetting ability and bonding between them. The extreme reaction between particles and matrix may have an undesirable impact on the mechanical and thermal properties of the composite, while a severe reaction can damage the reinforcements [25].

Thereby, an ideal reaction is desired for composite fabrication. Magnetite is commonly found in self-sustaining thermite reaction. The Al–Fe_3_O_4_ system is identified as a highly exothermic reaction that can be employed during mechanical or thermal treatments, based on the following stoichiometric reaction:3Fe_3_O_4_ + 8Al → 4Al_2_O_3_ + 9Fe   ΔH° = −3021 kJ(3)

One of the weak interface outcomes is reduction in stiffness, hardness, and strength, but high resistant to fractures. In contrast, strong interface between particles in the matrix shows high stiffness and strength, but typically low resistance to fracture [26,27]. Mechanical properties of the hybrid composite depend on the percentage of reinforcement materials, and microstructure and volume fraction of dendritic α-Al [28]. Adding silicon carbide has reformed the microstructure of Al-Fe_3_O_4_ composites, which improved mechanical properties.

In conformity with Ellingham–Richardson diagram, an oxygen-and-aluminum reaction is more likely to occur compared to iron and aluminum, so Fe_3_O_4_ would be reduced by aluminum. Notwithstanding, due to the negative free energy of creation for various Al-Fe intermetallic between aluminum and iron, there is a thermodynamic tendency to react with each other and form Al-Fe intermetallic compounds (IMCs) [29,30]. Based on binary phase diagram of Al-Fe, two main phases of Al_5_Fe_2_ and Al_3_Fe were recognized at the interface of iron and aluminum, which have high hardness (Al_3_Fe = ~717 Hv, Al5Fe_2_ = ~944 Hv) [31]. Furthermore, there are various Al-Fe intermetallic compounds such as AlFe, Al_2_Fe, Al_5_Fe_2_, AlFe_3_, and Al_3_Fe based on the Al-Fe phase diagram [30]. However, at temperatures above 550 °C, Al_3_Fe is the only stable phase, since the sintering process occurred at 600 °C.

The practical paths that the Al–Fe_3_O_4_ reaction would proceed with are as follows: (1) direct reaction of Fe_3_O_4_ to form Fe or (2) reduction of Fe_3_O_4_ through an intermediate reaction to form FeO, and then reduce to Fe. The reactions can be addressed as below [32]:Fe_3_O_4_ → 3FeO + 1∕2O_2_(4)
2Al + 3FeO → Al_2_O_3_ + 3Fe(5)
3Al + Fe → Al_3_Fe(6)

For the Al-Fe-Si composition, at a temperature above 600 °C, Al_2_Fe_3_Si_4_ is at a stable phase, according to the previous study by Raghavan [33]. Figure 2 shows FESEM micrographs of Al-15Fe_3_O_4_ composite with uniform distribution Fe_3_O_4_ powders in different magnifications.

The EDS analysis results from four areas of Al-15Fe_3_O_4_ surface sample are shown in Figure 3 and were based on Figure 2. EDS analysis of the exposed surface point 1 shows the result for aluminum with 85.63 wt%. In point 2, an Fe_3_O_4_ particle (white color) exists in the aluminum matrix, with confirmed peaks of Fe (27.73 wt%) and O (31.63 wt%). Based on the EDS analysis, it indicates that the intermetallic phase of Al_3_Fe and interfaces of Al_2_O_3_ occurred at selected points of 3 and 4 with Al 70.11 wt%, Fe 27.73 wt%, and Al 85.65 wt% detected, respectively, which were also confirmed in to be in accordance with the XRD analysis.

The FESEM micrograph, in different magnifications for Al-30Fe_3_O_4_-20SiC, is also shown in Figure 4 composites after sintering at 600 °C and being etched. The Al matrix with homogeneous distribution of Fe_3_O_4_ powders (bright) and SiC particles (dark) in rectangular shape was positioned at the grain boundaries. Moreover, homogeneous distribution of particles in the matrix is evident. By increasing the amount of weight percentage of reinforcements, the possibility of agglomeration at grain boundaries is also increased. Figure 4c,d reveals a fully uniform distribution of particles, without any evidence of particle clustering.

The EDX analysis results from six areas of Al-30Fe_3_O_4_-20SiC sample surfaces, which are shown in Figure 4, are presented in Figure 5. The EDX results show Al, Fe, Si, O, and Mg in the grain boundaries. Point 1 is aluminum, where Al peaks (82.83 wt%) are the main peak, and it is detected by using FESEM, in a light gray color. Point 2 is Fe_3_O_4_ (Fe 33.55 and O 42.35 wt%) in white particles, and point 3 is confirmed as SiC (Si 46.75 and C 10.7 wt%) in a dark gray color, and they were also confirmed by XRD pattern. According to the EDS analyses, the compositions of Al_3_Fe (Al 57.76 and Fe 18.84 wt%), Al_2_O_3_ (Al 29.10 and O 45.15 wt%), and Al_2_Fe_3_Si_4_ (Al 20.5, Fe 30.76, and Si 41.25 wt%) at number 4, 5, and 6 were detected, respectively, which were also confirmed by XRD pattern. This analysis is also in agreement with previous researches [34,35,36].

By comparing microstructure images for Al-Fe_3_O_4_ and Al-Fe_3_O_4_-SiC composite in Figure 2 and Figure 4, we conclude that the addition of silicon carbide modified the microstructure of the Al-Fe_3_O_4_ composite. As shown in Figure 2, the proeutectic plates (Fe_3_O_4_) are more rectangular in shape, while, in Figure 4, by adding silicon carbide, the microstructure has evolved from rectangular to a more spherical shape.

### 3.2. Characterization XRD

The XRD analysis in Figure 6 displays the phase identification in the specimens. The X-ray diffraction of two composites Al-15Fe_3_O_4_ analyses and Al-30Fe_3_O_4_-20SiC after sintering were shown. The measurements of X-ray diffraction, after the addition of reinforcements, were studied by comparing the peaks with diffraction. As can be seen, the diffraction peak with the highest intensity is related to aluminum. Al-Fe intermetallic compound (Al_3_Fe, gray platelets) and Al_2_O_3_ were detected in Al-15Fe_3_O_4_ XRD analyses. In Figure 6, for Al-30Fe_3_O_4_-20SiC the first peak assigned to SiC (Hexagonal), and the second peak were referred to as Fe_3_O_4_ with a cubic crystallography system. Moreover, minor amounts of Al-Fe intermetallic compounds (Al_5_Fe_2_-Al_3_Fe), Fe_3_C cementite (iron carbide), and Al_2_Fe_3_Si_4_ iron aluminum silicon were identified. After adding hybrid reinforcement particles (Fe_3_O_4_-SiC) and heat-treating, 2θ = (38/784, 44/600, 65/186, 78/306, 82/352) is related to aluminum, 2θ = (35/439, 43/070, 62/546) is assigned to the Fe_3_O_4_ cubic crystal system, and 2θ = (35/731, 59/996, 71/944) is related to the SiC cubic crystal system. Moreover, Al_2_O_3_ was detected.

### 3.3. Density Measurements

The density of magnetite Fe_3_O_4_ (4.8 g/cm^3^) is much higher than aluminum (2.70 g/cm^3^) and SiC (3.21 g/cm^3^). The density of Al-Fe_3_O_4_ and Al-Fe_3_O_4_-SiC hybrid composites, both before and after sintering, was assessed by using the Archimedes’ principle [37]. The eight samples’ density in Figure 7 varies from 2.37 for Al-15Fe_3_O_4_ before sintering to 2.72 g/cm^3^ for Al-30Fe_3_O_4_-15SiC after sintering. The density of composites was measured, using the water-displacement method, and the theoretical density of composites was calculated by rule of mixture equations. By comparing the theoretical density values, the density of composite is acceptable, showing the samples as fully dense. Since the density of Fe_3_O_4_ is higher than SiC, the combination with higher weight percentage of (Fe_3_O_4_-SiC) has shown the highest amount of density. By increasing the weight percentage of hybrid fillers, the density of composite also increases. The compact pressure for all compositions was kept at constant 250 MPa.

### 3.4. Hardness Test

In Figure 8, the micro-hardness of the hybrid composites was measured five times, and the average results were presented. The mechanical properties of the hybrid composites depend on the interaction between the particles and matrix, microstructure, sintering temperature, and weight percentage of reinforcement materials [38]. In accordance, the different weight percentages of hybrid nanoparticles reinforcements added into aluminum matrix and an optimum amount of hybrid reinforcement required to modify the mechanical properties of HAMC were investigated. In Figure 8, by increasing the amount of reinforcement (wt% of SiC), we improved the hardness.

The development in the hardening of the composite can contribute to the higher stiffness of silicon carbide particles, as well as strong interfacial bonding between Al and SiC. Generally, the addition of ceramic nanoparticles stopped the movement of dislocations, which limited the deformation of the nanocomposite, which is the main reason in determining the increase of micro hardness in the HAMC. The various weight percentages of hybrid fillers were added into the aluminum matrix, and an optimum amount of hybrid filer (SiC-Fe_3_O_4_) was determined, to optimize the mechanical properties. Figure 8 displays the samples’ hardness values, depending on the percentage of reinforcements in the matrix from 44 HV for Al-15Fe_3_O_4_ to 93 HV for Al-15Fe_3_O_4_-30SiC. By increasing the weight percentage of silicon carbide, the micro-hardness of the composite had increased dramatically and improve the hardness value.

Adding 30 wt% of SiC to the Al-15Fe_3_O_4_ composite improved the hardness values of specimen by 111%. However, by rising the amount of iron oxide to 30 wt% with different amounts of SiC in the composite, the micro-hardness value was also increased from 46 to 91 HV. By increasing the weight percentage of silicon carbide, the micro-hardness of the composite increased intensely, while raising the amount of iron oxide slightly increased the hardness of the composite.

### 3.5. Tribology Analysis

Usually, while doing wear analysis, the structure and characteristics of the near-surface areas and the surface can be modified. Materials which split into two sliding surfaces can be as a distinct “third-body” with their own properties. These characteristics will regularly be altered throughout the system’s lifetime. Furthermore, the surface topography, while doing a wear test, might be modified due to the elimination or displacement of the materials. In contrast, the identification of wear mechanisms is usually multifaceted, since it involves a combination of mechanical and chemical processes [39]. In this experiment, AFM imaging was utilized to evaluate the topographical properties of the worn and plain surfaces. When we applied the same load, the wear indicated various upshots on the surface, where various wear modes were detected. By observing the depths and widths of wear grooves, diverse deductions can be done which are explained in detail in the following paragraphs.

In Figure 9, the topographic images of the undamaged and wear surfaces on the Al-15Fe_3_O_4_ and Al-15Fe_3_O_4_-30SiC samples are presented. From Figure 9a, some surface defects produced by the manufacturing process are evident on each sample. These surface defects may remain intact after polishing, cleaning, and sonication steps, but their remains are related to the surface deficiencies. In Figure 9b, the occurrence of coarse ridges and grooves caused by the intense plastic deformation is the crucial feature of the worn surface. This result is consistent with previous studies, which confirmed that the wear happened due to homogeneous deformation in isothermal mode and through which the material was detached as a result of the lip formation and extrusion [40]. Normally, the onset of cracks on the surface is the cause of debris formation in the transfer layer. With this consideration to the Al-Fe_3_O_4_, the fracture toughness, rather than hardness, is the key mechanical feature to control the wear rate. Consequently, the Al-15Fe_3_O_4_ with high resistance to the disconnection and pull-out inhibits the adhesive wear. The AFM images of samples were gathered. The Al-15Fe_3_O_4_-30SiC reveals an almost-dense structure (Figure 9c). By adding SiC reinforcement, we lowered the amount of wear, compared to the Al-15Fe_3_O_4_ (Figure 9d). From the sample configuration, the Al-15Fe_3_O_4_ surface topography was characterized by a homogenous structure containing dimples and hillocks. In Figure 9c, it is indicated that the plastic deformation level in terms of ploughs and transverse tracks was observed to be lower when compared to other samples.

Surface roughness is one of the main terms in the material removal process which refers to the finely spaced surface irregularities produced by the machining operation in the case of machined surfaces. Roughness can be measured by the height of the irregularities with respect to an average line and is usually stated in micrometers or micro-inches [39]. Here, the roughness average (*R_a_*; the arithmetic average of absolute values) was used, and it is defined as follows:(7)$$R_{a}=\frac{1}{n}\sum_{i=1}^{n}\left|y_{i}\right|$$
where *y_i_* and *n* are the vertical distance from the centerline and the total number of vertical measurements taken within a specified cutoff distance, respectively.

Based on the achieved data, the *R_a_* amount is approximately 701, 1207, 349.9, and 476 nm for the Al-15Fe_3_O_4_ and Al-15Fe_3_O_4_-30SiC, before and after wear, respectively. It is concluded that the surface of Al-15Fe_3_O_4_-30SiC is not smooth, while the roughness becomes lower when compared to the Al-15Fe_3_O_4_. As it can be seen, the *R_a_* amount inside the wear tracks of the Al-15Fe_3_O_4_-30SiC samples is approximately 476 nm, almost 61%, lower than the value measured for the Al-15Fe_3_O_4_ (1207 nm). This shows the essential role of SiC in reducing wear, which is crucial for different applications.

A strong interfacial interaction between Al matrix and SiC with strong Al–Si bonding is the reason for higher wear resistance in the composite. In fact, Fe_3_O_4_-SiC particles with higher strength than aluminum can withstand the applied load without much deformation. Thus, if the particles distribute homogeneously, they can improve the wear properties. Moreover, the Al_3_Fe intermetallic compound reduces formability and fatigue resistance, but improves wear resistance. The presence of the intermetallic compound with high micro-hardness will improve the hardness value and wear resistance [41].

Figure 10 indicates the coefficient of friction (COF) versus cumulative sliding time for the Al-15Fe_3_O_4_, Al-30 Fe_3_O_4_, Al-15Fe_3_O_4_-30SiC, and Al-30Fe_3_O_4_-20SiC specimens under normal loads of 10 N. For all cases, a common feature was investigated with the constant load. Based on the graph, the coefficient of friction increases abruptly and reached a steady state in a short time interval. This phenomenon is cause by the consecutive wear of surface asperities and improved compliance of smooth worn surfaces. Therefore, wear is due to the brittle micro-fractures in the surface grains and tribo-chemical reaction in the initial and final stages, respectively [42]. Furthermore, it is obvious that the quick improvement in the COF is due to added reinforcements (SiC and Fe_3_O_4_) to the composite by powder metallurgy method. The plotted friction lines reach a steady state within the initial few seconds, signifying that each sample has the same wear pattern for the applied load. The COF increases significantly in the constant load of 10 N and reveals the COF values in the range of 0.412 to 0.601. It should be declared that the expected decrease in the COF value from Al-15Fe_3_O_4_ to Al-30Fe_3_O_4_ is related to increasing Fe_3_O_4_ from 15 to 30 wt%. In addition, Al-15Fe_3_O_4_-30 SiC has nearly the same COF value as Al-30Fe_3_O_4_, which is an effect of adding 30 wt% SiC to the composite without increasing Fe_3_O_4_ from 15 to 30 wt%. In addition to this finding, the COF values of the Al-30Fe_3_O_4_-20Sic specimens declined dramatically in the 10 N applied load, as compared to Al-15Fe_3_O_4_-30SiC, which is related to the added Fe_3_O_4_.

### 3.6. Effectiveness of Corrosion Protection

To study the corrosion and passivation kinetics of the samples, we utilized the potentiodynamic polarization method (Figure 11). This experiment was performed on the Al-15Fe_3_O_4_, Al-15Fe_3_O_4_-30SiC, Al-30Fe_3_O_4_, and Al-30Fe_3_O_4_-20SiC specimens which were exposed to artificial seawater medium. The results of corrosion current density (*I*_corr_), corrosion potential (*E*_corr_) and corrosion rate are summarized in Table 2. *E*_corr_ represents the tendency for the substrate to corrode, whilst *I*_corr_ represents the efficiency of corrosion protection. As seen from Table 2, the *E*_corr_ values falls within the range of −615 to −729 mV for all the tested samples. The *I*_corr_ value of Al 15 Fe_3_O_4_ is 6.157 × 10^−4^ μA; meanwhile, by increasing wt% Fe_3_O_4_ to 30% (Al 30 Fe_3_O_4_), it is improved to 7.342 × 10^−5^ μA. The values are then further decreased by adding SiC to 5.591 × 10^−5^ μA for Al 15 Fe_3_O_4_ 30 Sic and finally 1.018 × 10^−6^ μA for the Al-30Fe_3_O_4_-20SiC sample.

By calculating the corrosion current, the equivalent weight and density for each sample corrosion rate are obtained [43,44]. Corrosion rate for Al-15Fe_3_O_4_ shows 8.4 × 10^−4^, while for Al-30Fe_3_O_4_, the corrosion rate decreased significantly to 0.87 × 10^−4^. This explains that the addition of wt% Fe_3_O_4_ helps to minimize the corrosion rate in a corrosive environment. Moreover, the corrosion rate for Al-15Fe_3_O_4_-30SiC shows an even lower rate, which is 0.8 × 10^−4^, and the rate decreases to 0.01 × 10^−4^ for Al-30Fe_3_O_4_-20SiC which is near zero and it is concluded that this composition is the optimum composite. Furthermore, as expected, the corrosion protection efficiency increases from 88.07%, 90.91%, and 99.83% for Al-30Fe_3_O_4,_ Al-15Fe_3_O_4_-30SiC, and Al-30Fe_3_O_4_-20SiC samples, respectively. Therefore, adding and optimizing these reinforcements (Al-Fe_3_O_4_-SiC) to the composite shows a positive outcome towards corrosion resistance to increase the durability and life span of material during operation.

To further explain the better corrosion resistance of the hybrid composite at room temperature compared with the pure Al matrix, we note that SiC particulates are ceramics and remain inactive in the artificial seawater solution. They are barely influenced by the artificial seawater medium. According to previous researches, the corrosion resistance of the aluminum matrix composite is controlled by many elements, such as the fabrication technique, characteristics of the matrix, the amount of the reinforcement, and the environmental features. Formation of Al_4_C_3_ at the reinforcement/matrix interface has a negative effect on corrosion, while the undesirable Al_4_C_3_ has not been formed base on the XRD analysis. Accordingly, the formation of Al_4_C_3_, Al_2_Fe_3_Si_4_, iron, aluminum, or silicon phase at the SiC particles/matrix interface during fabrication did not happen in this experiment. It is assumed that the SiC particles play a main role as a physical barrier. The particles acts as a firm barrier to the formation of corrosion pits [45]. The weakest part of composites is the interface between the matrix and the reinforcement. Thereby, the interfacial bond, strong or weak, is important in the corrosion process [46].

Moreover, the observations revealed that Fe_3_O_4_ reinforced composites have higher corrosion resistance, compared to the aluminum matrix. Note that the detected pitting corrosion in aluminum base metal was not discovered in the existence of Fe_3_O_4_ nanoparticles. This might be in consequence of the existence of Al_3_Fe and Al_5_Fe_2_ intermetallics, which serve as cathodes with regard to the metal matrix and enhance pitting corrosion resistance. The developed pitting behavior is more essential in the presence of chloride ions, while the aluminum surface becomes very vulnerable to pitting corrosion. Thus, it can be determined that Fe_3_O_4_ nanoparticles had a dual influence on the corrosion resistance of the aluminum matrix composite [47]. Based on the obtained results, it can be deduced that Al_2_O_3_ improves the corrosion resistance because Al_2_O_3_ affects the anodic and cathodic reactions [48].

## 4. Conclusions

Novel and optimized Al-Fe_3_O_4_-Sic composites were successfully prepared to improve mechanical, tribological, and corrosion properties in order to identify the optimum amount of filler without mechanical degradation, using low-cost powder metallurgy method. Based on the obtained results and carried out experiments and discussions, adding hybrid reinforcement particles into aluminum made it multifunctional low-weight materials by developing magnetic properties. The distribution of hybrid reinforcement particles in the AMC were homogeneous, and positively impact the surface hardness of aluminum by 111%. The results of tribology showed that COF value in the 10 N applied load decreased from 0.601 for Al-15Fe_3_O_4_ to 0.412 for Al-30Fe_3_O_4_-20 SiC, which is related to the adding of Fe_3_O_4_ and SiC reinforcement to the composite. After AFM assessment, the *R_a_* values inside the wear tracks of the Al-15Fe_3_O_4_ 30 Sic samples were nearly 476 nm, almost 61% lower than the value measured for the Al15 Fe_3_O_4_. Finally, the corrosion results demonstrated that Al-30Fe_3_O_4_-20 SiC led to a significant improvement in *P.E.* (99.83%). By comparing all samples, we believe Al-30Fe_3_O_4_-20SiC can be selected as an optimization composite.

## Figures and Tables

**Figure 1 materials-13-04090-f001:**
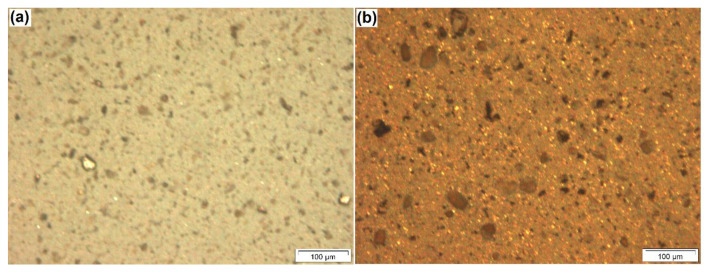
Optical microscopy for two compositions: (**a**) Al-15 Fe_3_O_4_ and (**b**) Al-30 Fe_3_O_4_-20SiC.

**Figure 2 materials-13-04090-f002:**
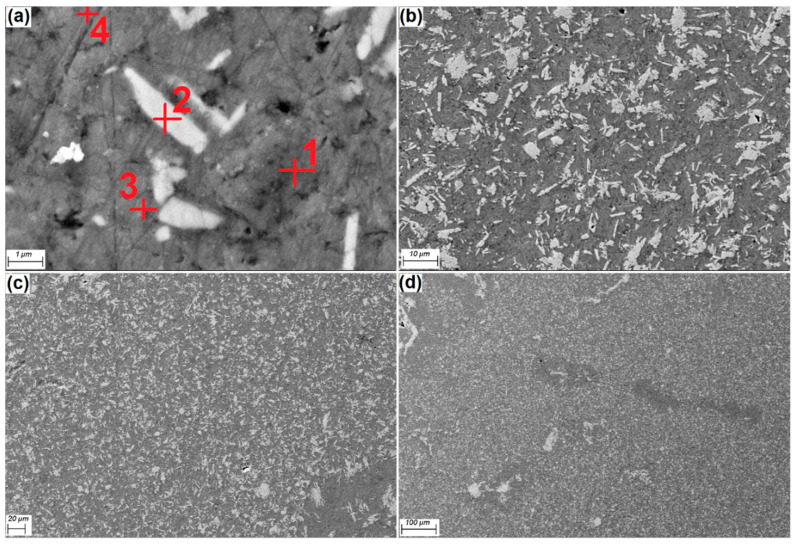
FE-SEM micrographs of Al-15Fe_3_O_4_ composite in different magnifications (**a**) 1 μm scale, (**b**) 10 μm scale, (**c**) 20 μm scale and (**d**) 100 μm scale.

**Figure 3 materials-13-04090-f003:**
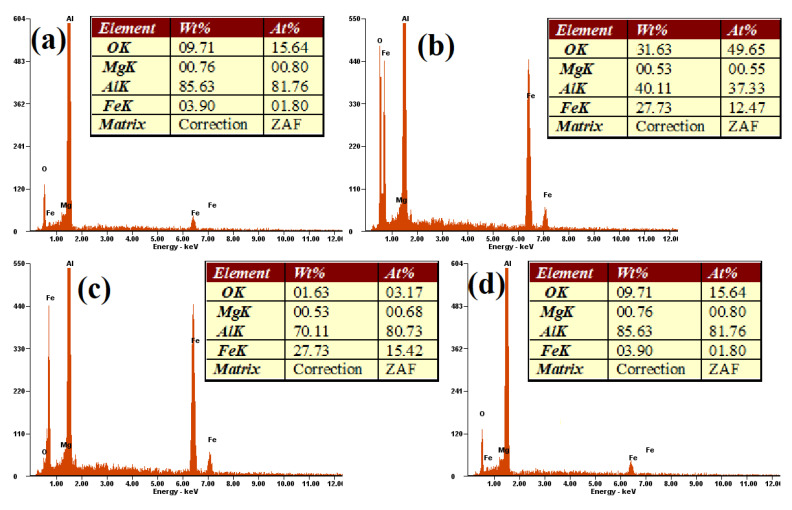
The EDS spectrum of Al-15Fe_3_O_4_ composition: (**a**) aluminum, (**b**) Fe_3_O_4_, (**c**) Al_3_Fe, and (**d**) Al_2_O_3_.

**Figure 4 materials-13-04090-f004:**
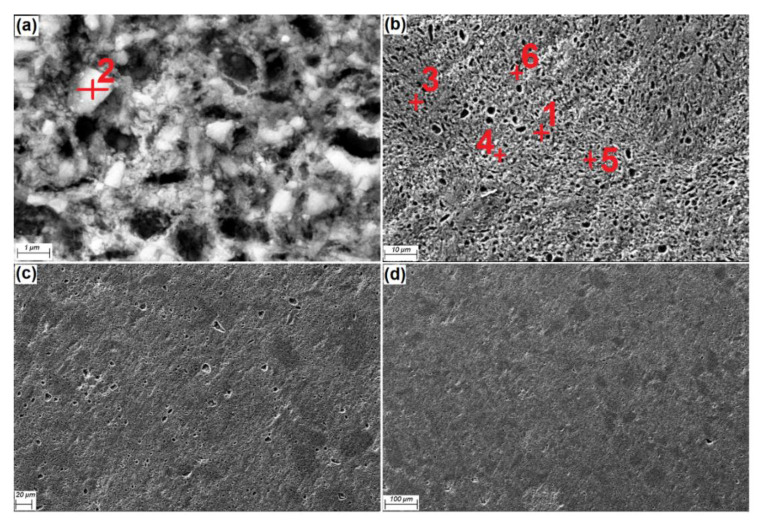
FESEM micrographs of Al-30Fe_3_O_4_-20SiC composite, in different magnifications, after etching (**a**) 1 μm scale, (**b**) 10 μm scale, (**c**) 20 μm scale and (**d**) 100 μm scale.

**Figure 5 materials-13-04090-f005:**
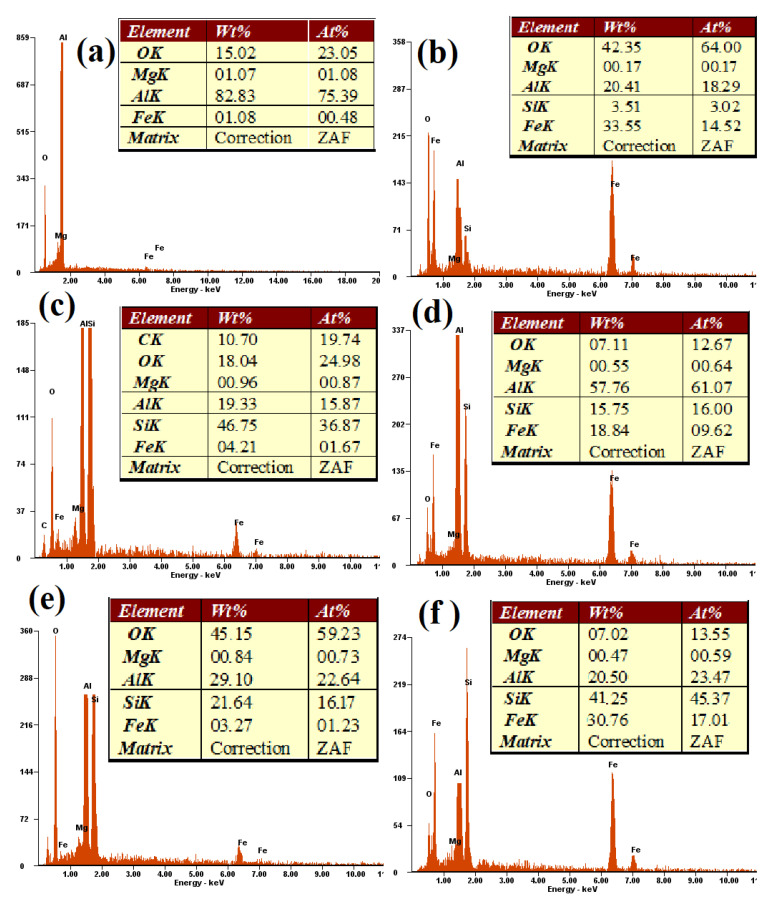
The EDS spectrum of Al-30Fe_3_O_4_-20SiC composition: (**a**) aluminum, (**b**) Fe_3_O_4_, (**c**) SiC, (**d**) Al_3_Fe, (**e**) Al_2_O_3_, and (**f**) Al_2_Fe_3_Si_4_.

**Figure 6 materials-13-04090-f006:**
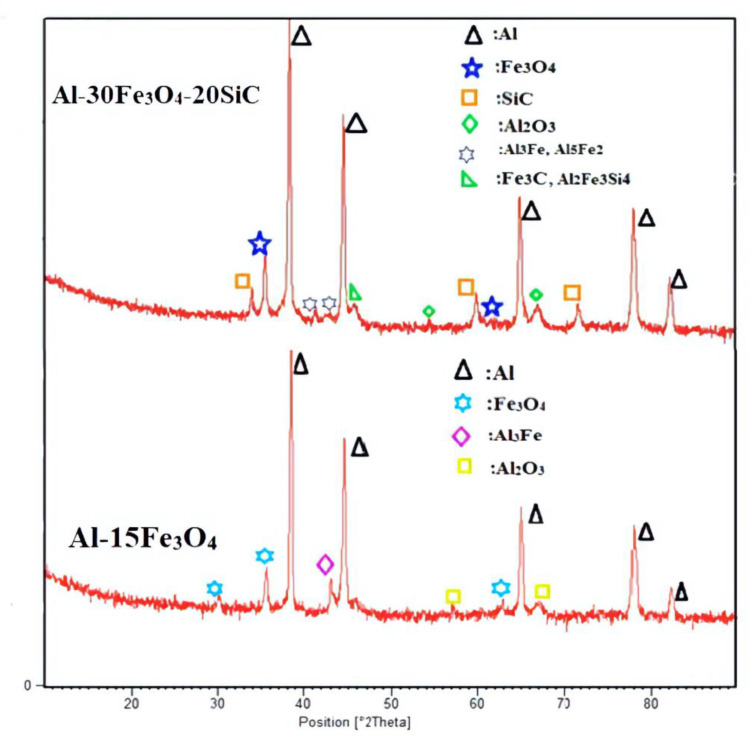
Shows the X-ray diffraction (XRD) analyses for the two composites.

**Figure 7 materials-13-04090-f007:**
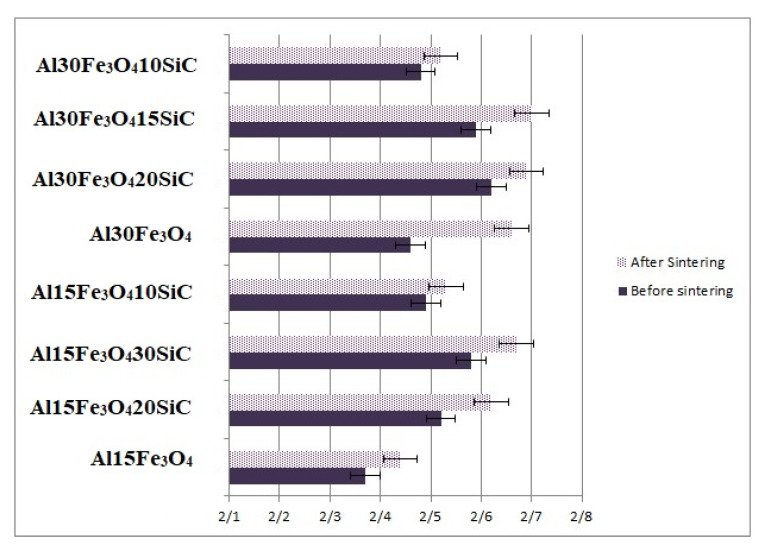
Evolution of density, both after and before sintering, depending on hybrid filler.

**Figure 8 materials-13-04090-f008:**
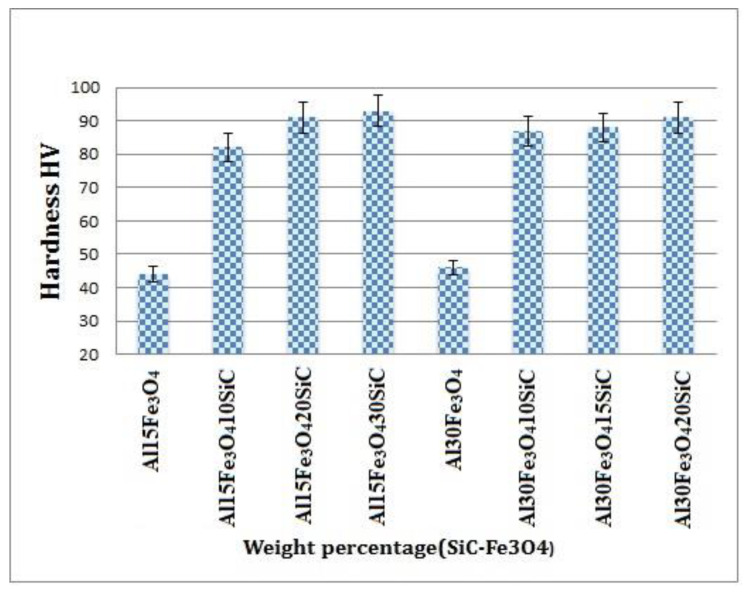
The variation of Vickers hardness value of the as-prepared different composites of Al-Fe_3_O_4_-SiC.

**Figure 9 materials-13-04090-f009:**
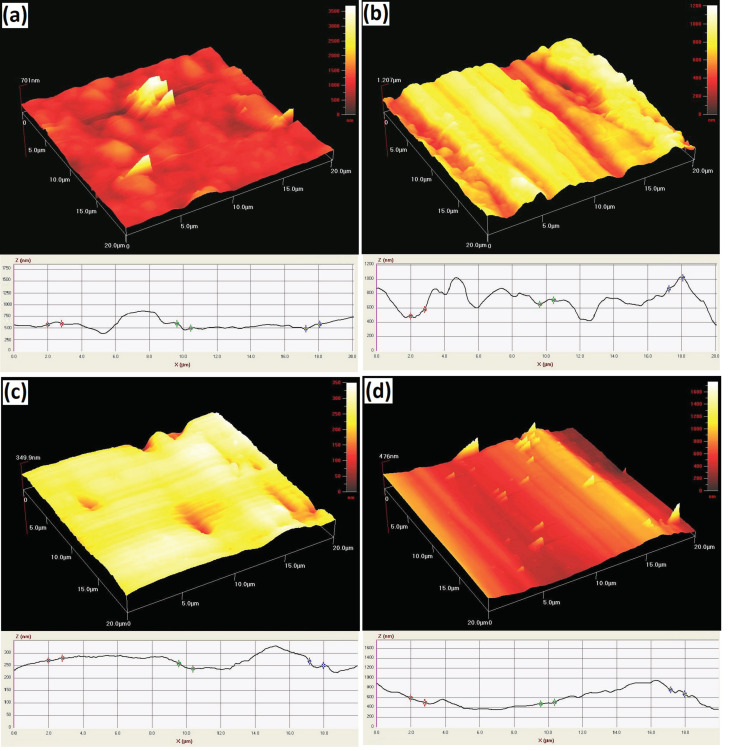
Topographic images of undamaged and wear surfaces on (**a**,**b**) Al-15Fe_3_O_4_ and (**c**,**d**) Al-15Fe_3_O_4_-30SiC specimens over an area of 20 μm × 20 μm.

**Figure 10 materials-13-04090-f010:**
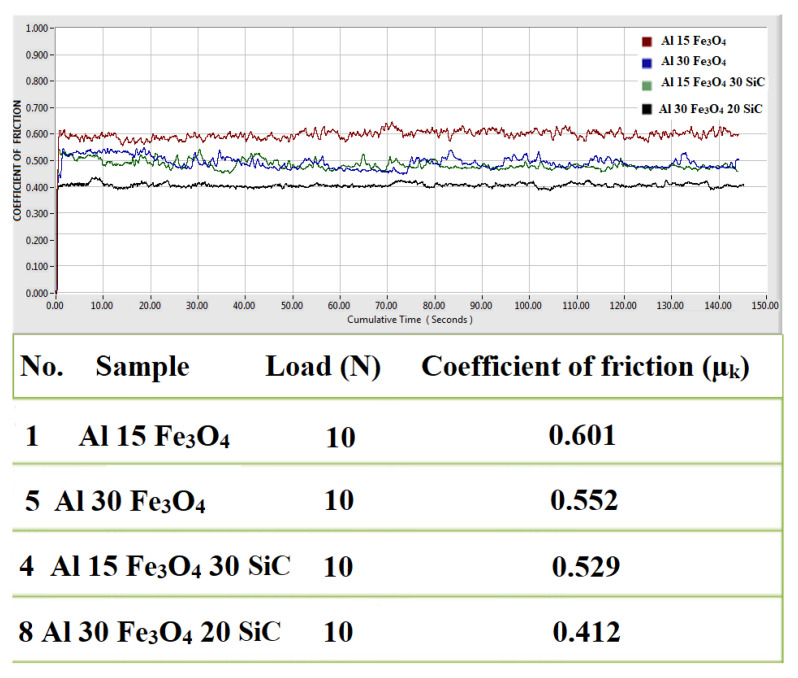
The coefficient of friction (COF) vs. cumulative sliding time for the Al-15 Fe_3_O_4_, Al-15Fe_3_O_4_-30SiC, Al-30Fe_3_O_4_, and Al-30Fe_3_O_4_-20SiC specimen.

**Figure 11 materials-13-04090-f011:**
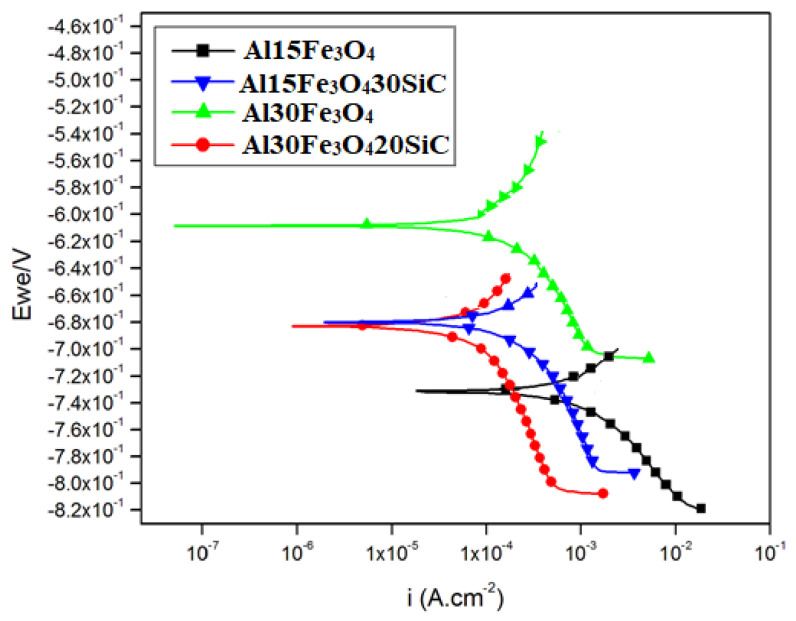
Polarization curves of Al-15Fe_3_O_4_, Al-15Fe_3_O_4_-30SiC, Al-30Fe_3_O_4_, and Al-30 Fe_3_O_4_-20SiC specimen in artificial seawater.

**Table 1 materials-13-04090-t001:** Different compositions of aluminum, ferrous ferric oxide, and silicon carbide.

Composition	Al (wt%)	Fe_3_O_4_ (wt%)	SiC (wt%)
sample 1	80	15	0
sample 2	70	15	10
sample 3	60	15	20
sample 4	50	15	30
sample 5	65	30	0
sample 6	55	30	10
sample 7	50	30	15
sample 8	45	30	20

**Table 2 materials-13-04090-t002:** Corrosion current density (*I*corr), corrosion potential (*E*corr), polarization resistance (*Rp*), corrosion rate, and effectiveness of corrosion protection (*P.E.*) data.

Parameter	Al-15Fe_3_O_4_	Al-30Fe_3_O_4_	Al-15Fe_3_O_4_-30 SiC	Al-30Fe_3_O_4_-20 SiC
Ecorr/mV	−729.522	−615.712	−680.156	−685.751
Icorr/A log (|<I>/A|)	6.157 × 10^−4^	7.342 × 10^−5^	5.591 × 10^−5^	1.018 × 10^−6^
Sample weights (gr)	2.21	2.75	2.42	2.91
Corrosion rate (mm/year)	8.4 × 10^−4^	0.87 × 10^−4^	0.8 × 10^−4^	0.01 × 10^−4^
*P.E.* (%)	-	88.07	90.91	99.83

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
