# Peer review of "Microstructural, Tribology and Corrosion Properties of Optimized Fe_3_O_4_-SiC Reinforced Aluminum Matrix Hybrid Nano Filler Composite Fabricated through Powder Metallurgy Method"

_materials, 2020, doi:10.3390/ma13184090_

Round 1
Reviewer 1 Report
1 Please unify the writing order of chemical equations.
2 What’s the CTE?
3 How does the wide band gap and high electron-mobility for SiC in this paper contribute to improvement of material properties.
4 For Fig 1 and 2, how to accurately determine the chemical composition of the material.
5 For Fig 5, the results of XRD do not match the above results.
6 In this paper, there are many selected components, but the results are not presented one by one.
7 Please strengthen the logic of the description of the experimental results.
8 What is the effect of reinforced particles on the corrosion properties of materials?
9 Please describe in further detail the effect of enhanced particles on friction and wear properties and corrosion resistance.
Author Response
Reviewer #1
1. Weakness of the manuscript is exact scientific terminology as well as English language. However, there are some discrepancy and missing technical details.
Thank you for this helpful comment. Action has been taken. We have sent the article to proof reading to improve the English language.
Technical Comments:
2. What is morphology of the source powder materials SiC and Fe3O4? What about morphology of ball milled Al-SiC-Fe3O4 composite powder?Thank you for your comment, due to current situation (coronavirus),the labs are closed and we cannot access to the FESEM lab. Although, we add the source powder materials (company name) and other details in the methodology.3. Compaction procedure is not clear defined. Is it cold isostatic pressing or die uniaxial cold pressing? What was pressing pressure? Is it really 250 Kgf? Similarly, sintering procedure is unclear described. Is it vacuum sintering and argon sintering? Is it provided in the same furnace? What about heating and cooling rate?Thank you for your helpful comment, action has been taken as follows:
Pure aluminium powder with a purity of 99.7 % with an average particle size of 20 μm was used as the composite matrix. Commercially available SiC with an average grain size of 2 μm and Fe3O4 (45-70 nm) purchased from (MHC Industrial Co., Ltd, China) were used to reinforce the Al matrix. The fabrication procedures were diligently carried out by firstly mixing the SiC and Fe3O4 powders and Al matrix particles were mechanically milled for 2 h using planetary ball mill (PM 100, Retsch, Haan, Germany) at a speed of 400 rpm at room temperature. The milling ball-to-powder weight ratio was at 15:1. Proper mixing is essential in the powder metallurgy process. The blended powder with a binder (Mg Stearate) was discharged into a tubular die (diameter of 20 mm). During ball milling, adding magnesium stearate can avoid agglomeration of particles and improve the distribution of reinforcements in the structure. Secondly, using universal testing machine (Instron 3382) compaction attains green compacts of powder, and then cold iso pressed (CIP) in one direction at a pressure of 2500 Kgf/cm for 15 minutes to attain an initial green density ranging from 85 to 95%. By using Linn High Therm furnace, the compacts were heated at 600 °C. Sintering process was done under argon atmosphere to prevent oxidation and temperature was fixed at 600°C for 20 minutes, with heating and cooling rate of 5°C/min, and then soaked in the furnace for 24 hours. Eight basic composition mixtures of magnetic nano iron oxide and silicon carbide were as presented in Table 1. All of the compositions comprise 5% Mg Stearate powder. For evaluation purpose of microstructural characterisation, specimens were prepared by grinding on various abrasive papers of 800, 1200, 2000 and 2500 grit, and polishing with diamond paste using alumina slur and ultrasonic cleaning in acetone and deionized water for 10 min, and drying at 100 °C for 1 h.
Results and discussion4. The results presented in microstructural evaluation and XRD characterization are inconsistent. Results of XRD analysis is presentation of Al2Fe3Si4, which is not noticed in SEM and LOM microstructure analysis. Thank you for your helpful comment, clarification has been made as follows:
3.1. Microstructural evaluation
The optical microscopy images for Al-15Fe3O4 and Al-30Fe3O4-20SiC composite are presented in Figure 1a, and 1b respectively. Figure 1a shows homogenous distribution of Fe3O4 particles in Al matrix. Figure 1.b is SiC as grey colour element, and Fe3O4 particles as white colour element which is distributed quite uniformly in Al-30Fe3O4-20SiC sample.
Figure 1: Optical microscopy for two composition of (a) Al-15 Fe3O4 (b) Al-30 Fe3O4-20SiC
Efficient reinforcement requires well bonded principal with matrix and particles. Chemical bonding (covalent, metallic, ionic) inter-diffusion is the diffusion of atoms between two metals, and van der Waals bonding are the components of the interface mechanisms pertaining to the filler and matrix bonding, and the reaction among the matrix and reinforcements in composite.
The appropriate reaction among matrix and reinforcements assists wetting ability and bonding between them. The extreme reaction between particles and matrix may have an undesirable impact on the mechanical and thermal properties of the composite, while a severe reaction can damage the reinforcements [25].
Thereby, an ideal reaction is desired for composite fabrication. Magnetite is commonly found in self-sustaining thermite reaction. The Al–Fe3O4system is identified as highly exothermic reaction that can be employed during mechanical or thermal treatments based on the following stoichiometric reaction:
3Fe3O4 + 8Al → 4Al2O3 + 9Fe ΔH° = −3021 kJ (3)
One of the weak interface outcomes is reduction in stiffness, hardness and strength but highly resistant to fractures. In contrast, strong interface between particles in the matrix shows high stiffness and strength but typically low resistant to fracture. [26,27].Mechanical properties of the hybrid composite depends on the percentage of reinforcement materials, and microstructure and volume fraction of dendritic α-Al [28]. Adding silicon carbide has reformed the microstructure of Al-Fe3O4 composites, which improved mechanical properties.
In conformity with Ellingham–Richardson diagram, oxygen and aluminium reaction is more likely to occur compared to iron and aluminium, so Fe3O4 would be reduced by aluminium. Notwithstanding, due to the negative free energy of creation for various Al-Fe intermetallic between aluminium and iron there is a thermodynamic tendency to react with each other and form Al-Fe intermetallic compounds (IMCs) [29,30]. Based on binary phase diagram of Al-Fe ,two main phases of Al5Fe2 and Al3Fe were recognized at the interface of iron and aluminium, which have high hardness (Al3Fe = ~717 Hv, Al5Fe2 = ~944 Hv) [31]. Furthermore, there are various Al-Fe intermetallic compounds such as AlFe, Al2Fe, Al5Fe2, AlFe3 and Al3Fe based on Al-Fe phase diagram [30]. Although at temperature above 550°C, Al3Fe is the only stable phase since the sintering process occurred at 600°C.
The practical paths that Al–Fe3O4 reaction would proceed with are: (1) direct reaction of Fe3O4 to form Fe or (2) reduction of Fe3O4 through an intermediate reaction to form FeO, and then reduce to Fe. The reactions can be addressed as below [32]:
Fe3O4 → 3FeO + 1∕2O2 (4)
2Al + 3FeO → Al2O3 + 3Fe (5)
3Al + Fe → Al3Fe (6)
The phase diagram of Al-Fe-Si ternary system at 600◦C is shown in Figure 2 which illustrates that Al2Fe3Si4 is a stable phase at temperature above 600◦C.
Figure 2:Al-Fe-Si Computed Phase Diagram at 600◦C [33].
Figure 3 shows FESEM micrographs of Al-15Fe3O4 composite with uniform distribution Fe3O4 powders in different magnifications.
Figure 3: FE-SEM micrographs of Al-15Fe3O4 composite in different magnifications
Figure 4: The EDS spectrum of Al-15Fe3O4 composition (1) Aluminuim (2) Fe3O4 (3) Al3Fe(4) Al2O3
The EDS analysis results from four areas of Al-15Fe3O4 surface sample are shown in Figure 4 which were based on Figure 3. EDS analysis of the exposed surface point 1 shows the result for Aluminium with 85.63wt%. In point 2, Fe3O4 particle (white colour) exists in aluminium matrix with confirmed peaks of Fe (27.73wt%) and O (31.63wt%). Based on EDS analysis, it indicates that intermetallic phase of Al3Fe and interfaces of Al2O3occurred at selected points of 3 and 4 with Al 70.11wt%, Fe 27.73wt% and Al 85.65wt% were detected, respectively which were also confirmed in accordance to XRD analysis.
Figure 5: FESEM micrographs of Al-30Fe3O4-20SiC composite in different magnifications after etching
FESEM micrograph in different magnifications for Al-30Fe3O4-20SiC is also shown in Figure 5 composites after sintering at 600 °C and etched. Al matrix with homogeneous distribution of Fe3O4 powders (bright) and SiC particles (dark) in rectangular shape were positioned at the grain boundaries. Moreover, homogeneous distribution of particles in the matrix is evident. By increasing the amount of weight percentage of reinforcements, the possibility of agglomeration at grain boundaries is also increased. Figure 5c and d reveals a fully uniform distribution of particles, without any evidence of particle clustering.
The EDX analysis results from six areas of Al-30Fe3O4-20SiC sample surfaces which are shown in Figure 5 are then presented in Figure 6. The EDX results show Al, Fe, Si, O and Mg in the grain boundaries. Point 1 is Aluminium where Al peaks (82.83wt%) are the main peak and it is detected using FESEM in light grey colour. Point 2 is Fe3O4 (Fe 33.55 and O 42.35 wt%) in white particles and Point 3 is confirmed as SiC (Si 46.75, and C 10.7 wt%) in dark grey colour which were also confirmed by XRD pattern. According to EDS analyses, composition of Al3Fe (Al 57.76 and Fe 18.84 wt%), Al2O3 (Al 29.10 and O 45.15 wt%), and Al2Fe3Si4 (Al 20.5, Fe 30.76 and Si 41.25 wt%) at number 4, 5 and 6 were detected, respectivelywhich were also confirmed by XRD pattern. This analysis is also in agreement with previous researches [34-36].
Figure 6: The EDS spectrum of Al-30Fe3O4-20SiC composition(1) Aluminium(2) Fe3O4(3)SiC(4) Al3Fe(5) Al2O3(6) Al2Fe3Si4
By comparing microstructure images for Al-Fe3O4 and Al-Fe3O4-SiC composite in Figures 3 and 5, it is concluded that the addition of silicon carbide had modified the microstructure of Al- Fe3O4 composite. As shown in Figure 3, proeutectic plates (Fe3O4) is more rectangular in shape, while in Figure 5, by adding silicon carbide, the microstructure has evolved from rectangular to a more spherical shape.
3.2. Characterization XRD
The XRD analysis in Figure 7 displays the phase identification in the specimens. The X-ray diffraction of two composites Al-15Fe3O4 analyses and Al-30Fe3O4-20SiC after sintering were shown. The measurements of X-ray diffraction after addition of reinforcements, were studied by comparing the peaks with diffraction. As can be seen, the diffraction peak with the highest intensity is related to Aluminium. Al-Fe intermetallic compound (Al3Fe, grey platelets) and Al2O3 were detected in Al-15Fe3O4 XRD analyses. In Figure 7, for Al-30Fe3O4-20SiC the first peak assigned to SiC (Hexagonal), and the second peak were referred to as Fe3O4 with a cubic crystallography system. Moreover, minor amount of Al-Fe intermetallic compounds (Al5Fe2 - Al3Fe), Fe3C cementite (iron carbide), Al2Fe3Si4 Iron Aluminium Silicon were identified. After adding hybrid reinforcement particles (Fe3O4-SiC), and heat treated, 2θ= (38/784, 44/600, 65/186, 78/306, 82/352) are related to aluminium, 2θ= (35/439, 43/070, 62/546) assigned to Fe3O4 cubic crystal system, and 2θ= (35/731, ,59/996, 71/944) related to SiC cubic crystal system. Moreover, Al2O3 has been detected.
Figure 7. Shows the X-ray diffraction (XRD) analyses for the two composites
5. Section 3.3 Density have to be rewritten. There is problematic sentence as “...samples have approximately the same 257 density...”. Fig.6 is necessary to correct in Y axes
Thank you for this comment. Action has been taken.
7. In 3.5 Tribology analysis, It is problematic to evaluate the results without missing tribo-pair information as well testing time or tribo-path is missing.
Thank you for this comment. Information has been added in methodology.
2.2.3 Wear Test
By utilizing a pin-on-disc configuration (Ducom Reciprocating Friction Monitor – TR 282 Series), the wear tests were performed in dry-sliding condition. This machine is intended to measure wear characteristics and the friction of the specimens, through reciprocating sliding movement. A reciprocating engine is utilized to generate a bi-directional sliding movement between the samples while a loading mechanism applies the chosen load upon the test samples. Moreover, the instant friction force by a friction measurement system can be measured. Coefficient of friction (COF) and a diversity of optional facilities are also measured and demonstrated on the "WinDucom" software. The dry-sliding experiment starts as the alumina cylindrical pin, in 6 mm diameter and 8 mm length, glides against a stationary counterpart plate. Before the wear test, both pins and samples were cleaned with distilled water and degreased with acetone. The normal loads of 10 N is kept constant while a reciprocating frequency of 10 Hz and amplitude stroke of 1 ± 0.02 mm were applied to the disc, where the tangential frictional force was continuously calculated using a load cell sensor attached to the pin-holder arm and recorded in a root mean square value. The kinetic coefficient of friction (μk) of each sample during 150 seconds duration was produced in the instrumentation output which was determined by dividing the recorded frictional force by the normal load. Besides, an atomic force microscope (AFM, Ambios Technology) was used to evaluate the topographical texture of the surfaces and wear scars (tribo-path).
8. Mechanical and corrosion properties have to be presented in relation to microstructure and XRD structure of the investigated material.
Thank you for this helpful comment.
For corrosion properties:
To further explain the better corrosion resistance of the hybrid composite at room temperature compared with the pure Al matrix is that SiC particulates are ceramics and remain inactive in the artificial sea water solution. They are barely influenced by the artificial sea water medium. According to previous researches the corrosion resistance of Aluminium matrix composite is controlled by many elements such as fabrication technique, characteristics of the matrix, the amount of the reinforcement and the environmental features. Formation of Al4C3 at the reinforcement/matrix interface has negative effect on corrosion, while the undesirable Al4C3 has not been formed due to XRD analysis. Accordingly, the formation of Al4C3, Al2Fe3Si4 Iron Aluminium Silicon, at the SiC particles/matrix interface during fabrication is unlikely to happen. It is assumed that the SiC particles play a main role as a physical barrier. The particles acts as a firm barrier to the formation of corrosion pits [45]. The weakest part of composites is the interface between the matrix and the reinforcement. Thereby, the interfacial bond, strong or weak, is important in the corrosion process [46].
Moreover, the observations revealed that Fe3O4 reinforced composites have higher corrosion resistance compared to aluminium matrix. Noteworthy, the detected pitting corrosion in aluminium base metal was not discovered in the existence of Fe3O4 nanoparticles. This might be in consequence of the existence of Al3Fe and Al5Fe2 intermetallic which serves as cathodes with regard to metal matrix and enhances pitting corrosion resistance. The developed pitting behaviour is more essential in the presence of chloride ions, while the aluminium surface becomes very vulnerable to pitting corrosion .Thus, it can be determined that Fe3O4 nanoparticles had a dual influence on the corrosion resistance of the aluminium matrix composite [47]. Based on the obtained results, it can be deduced that Al2O3 improves the corrosion resistance because Al2O3 affects the anodic and cathodic reactions [48].
For mechanical properties:
The development in the hardening of the composite can contribute to the higher stiffness of silicon carbide particles as well as strong interfacial bonding between Al and SiC. Generally, the addition of ceramic nanoparticles stopped the movement of dislocations which limited the deformation of the nanocomposite, which is the main reason in determining the increase of micro hardness in the HAMC.

Reviewer 2 Report
Topic of the article is interesting. It can be good contribution to knowledge in the field of Al matrix composites in connection with improving functional properties. Weakness of the manuscript is exact scientific terminology as well as English language. However, there are some discrepancy and missing technical details. What is morphology of the source powder materials SiC and Fe3O4? What about morphology of ball milled Al-SiC-Fe3O4 composite powder? Compaction procedure is not clear defined. Is it cold isostatic pressing or die uniaxial cold pressing? What was pressing pressure? Is it really 250 Kgf? Similarly, sintering procedure is unclear described. Is it vacuum sintering and argon sintering? Is it provided in the same furnace? What about heating and cooling rate?
Ad. Results and discussion
The results presented in microstructural evaluation and XRD characterization are inconsistent. Results of XRD analysis is presentation of Al2Fe3Si4, which is not noticed in SEM and LOM microstructure analysis. Section 3.3 Density have to be rewritten. There is problematic sentence as “...samples have approximately the same 257 density...”. Fig.6 is necessary to correct in Y axes. In 3.5 Tribology analysis, It is problematic to evaluate the results without missing tribo-pair informaition as well testing time or tribo-path is missing. Mechanical and corrosion properties have to be presented in relation to microstructure and XRD structure of the investigated material.
Author Response
Reviewer #2
1. Please unify the writing order of chemical equations.
Thank you for this comment. Action has been taken.
2. What’s the CTE? Thank you for this comment. Below item has been added to the manuscript. Coefficient of thermal expansion (CTE)3. How does the wide band gap and high electron-mobility for SiC in this paper contribute to improvement of material properties?
Thank you for this helpful comment. To explain more:
Silicon carbide have various excellent properties including low density, high strength, good high temperature strength (reaction bonded), oxidation resistance (reaction bonded), excellent thermal shock resistance, high hardness and wear resistance, excellent chemical resistance, low thermal expansion and high thermal conductivity. Wide-bandgap semiconductors permit devices to operate at much higher voltages, frequencies and temperatures compare to conventional semiconductor materials, The higher energy gap such as SiC gives devices the ability to operate at higher temperatures, as bandgaps typically shrink with increasing temperature, which can be problematic when using conventional semiconductors. For some applications, wide-bandgap materials allow devices to switch larger voltages. Moreover, by considering the properties of silicon carbide, wide its band gap can effect on electrical and thermal properties of the composite which is second part of the project and in this project investigated mechanical, tribology and corrosion behavior.
Reference
[1]Rosario Gerhardt, 2011, Properties and Applications of Silicon Carbide 4. For Fig 1 and 2, how to accurately determine the chemical composition of the material.Thank you for this helpful comment. Following literatures, EDX were used to determined the chemical composition of the material for FESEM and XRD to identify the compound. OM were also used to observe the cross sectional area.
3.1. Microstructural evaluation
The optical microscopy images for Al-15Fe3O4 and Al-30Fe3O4-20SiC composite are presented in Figure 1a, and 1b respectively. Figure 1a shows homogenous distribution of Fe3O4 particles in Al matrix. Figure 1.b is SiC as grey colour element, and Fe3O4 particles as white colour element which is distributed quite uniformly in Al-30Fe3O4-20SiC sample.
Figure 1: Optical microscopy for two composition of (a) Al-15 Fe3O4 (b) Al-30 Fe3O4-20SiC
Efficient reinforcement requires well bonded principal with matrix and particles. Chemical bonding (covalent, metallic, ionic) inter-diffusion is the diffusion of atoms between two metals, and van der Waals bonding are the components of the interface mechanisms pertaining to the filler and matrix bonding, and the reaction among the matrix and reinforcements in composite.
The appropriate reaction among matrix and reinforcements assists wetting ability and bonding between them. The extreme reaction between particles and matrix may have an undesirable impact on the mechanical and thermal properties of the composite, while a severe reaction can damage the reinforcements [25].
Thereby, an ideal reaction is desired for composite fabrication. Magnetite is commonly found in self-sustaining thermite reaction. The Al–Fe3O4system is identified as highly exothermic reaction that can be employed during mechanical or thermal treatments based on the following stoichiometric reaction:
3Fe3O4 + 8Al → 4Al2O3 + 9Fe ΔH° = −3021 kJ (3)
One of the weak interface outcomes is reduction in stiffness, hardness and strength but highly resistant to fractures. In contrast, strong interface between particles in the matrix shows high stiffness and strength but typically low resistant to fracture. [26,27].Mechanical properties of the hybrid composite depends on the percentage of reinforcement materials, and microstructure and volume fraction of dendritic α-Al [28]. Adding silicon carbide has reformed the microstructure of Al-Fe3O4 composites, which improved mechanical properties.
In conformity with Ellingham–Richardson diagram, oxygen and aluminium reaction is more likely to occur compared to iron and aluminium, so Fe3O4 would be reduced by aluminium. Notwithstanding, due to the negative free energy of creation for various Al-Fe intermetallic between aluminium and iron there is a thermodynamic tendency to react with each other and form Al-Fe intermetallic compounds (IMCs) [29,30]. Based on binary phase diagram of Al-Fe ,two main phases of Al5Fe2 and Al3Fe were recognized at the interface of iron and aluminium, which have high hardness (Al3Fe = ~717 Hv, Al5Fe2 = ~944 Hv) [31]. Furthermore, there are various Al-Fe intermetallic compounds such as AlFe, Al2Fe, Al5Fe2, AlFe3 and Al3Fe based on Al-Fe phase diagram [30]. Although at temperature above 550°C, Al3Fe is the only stable phase since the sintering process occurred at 600°C.
The practical paths that Al–Fe3O4 reaction would proceed with are: (1) direct reaction of Fe3O4 to form Fe or (2) reduction of Fe3O4 through an intermediate reaction to form FeO, and then reduce to Fe. The reactions can be addressed as below [32]:
Fe3O4 → 3FeO + 1∕2O2 (4)
2Al + 3FeO → Al2O3 + 3Fe (5)
3Al + Fe → Al3Fe (6)
The phase diagram of Al-Fe-Si ternary system at 600◦C is shown in Figure 2 which illustrates that Al2Fe3Si4 is a stable phase at temperature above 600◦C.
Figure 2:Al-Fe-Si Computed Phase Diagram at 600◦C [33].
Figure 3 shows FESEM micrographs of Al-15Fe3O4 composite with uniform distribution Fe3O4 powders in different magnifications.
Figure 3: FE-SEM micrographs of Al-15Fe3O4 composite in different magnifications
Figure 4: The EDS spectrum of Al-15Fe3O4 composition (1) Aluminuim (2) Fe3O4 (3) Al3Fe(4) Al2O3
The EDS analysis results from four areas of Al-15Fe3O4 surface sample are shown in Figure 4 which were based on Figure 3. EDS analysis of the exposed surface point 1 shows the result for Aluminium with 85.63wt%. In point 2, Fe3O4 particle (white colour) exists in aluminium matrix with confirmed peaks of Fe (27.73wt%) and O (31.63wt%). Based on EDS analysis, it indicates that intermetallic phase of Al3Fe and interfaces of Al2O3occurred at selected points of 3 and 4 with Al 70.11wt%, Fe 27.73wt% and Al 85.65wt% were detected, respectively which were also confirmed in accordance to XRD analysis.
Figure 5: FESEM micrographs of Al-30Fe3O4-20SiC composite in different magnifications after etching
FESEM micrograph in different magnifications for Al-30Fe3O4-20SiC is also shown in Figure 5 composites after sintering at 600 °C and etched. Al matrix with homogeneous distribution of Fe3O4 powders (bright) and SiC particles (dark) in rectangular shape were positioned at the grain boundaries. Moreover, homogeneous distribution of particles in the matrix is evident. By increasing the amount of weight percentage of reinforcements, the possibility of agglomeration at grain boundaries is also increased. Figure 5c and d reveals a fully uniform distribution of particles, without any evidence of particle clustering.
The EDX analysis results from six areas of Al-30Fe3O4-20SiC sample surfaces which are shown in Figure 5 are then presented in Figure 6. The EDX results show Al, Fe, Si, O and Mg in the grain boundaries. Point 1 is Aluminium where Al peaks (82.83wt%) are the main peak and it is detected using FESEM in light grey colour. Point 2 is Fe3O4 (Fe 33.55 and O 42.35 wt%) in white particles and Point 3 is confirmed as SiC (Si 46.75, and C 10.7 wt%) in dark grey colour which were also confirmed by XRD pattern. According to EDS analyses, composition of Al3Fe (Al 57.76 and Fe 18.84 wt%), Al2O3 (Al 29.10 and O 45.15 wt%), and Al2Fe3Si4 (Al 20.5, Fe 30.76 and Si 41.25 wt%) at number 4, 5 and 6 were detected, respectivelywhich were also confirmed by XRD pattern. This analysis is also in agreement with previous researches [34-36].
Figure 6: The EDS spectrum of Al-30Fe3O4-20SiC composition(1) Aluminium(2) Fe3O4(3)SiC(4) Al3Fe(5) Al2O3(6) Al2Fe3Si4
By comparing microstructure images for Al-Fe3O4 and Al-Fe3O4-SiC composite in Figures 3 and 5, it is concluded that the addition of silicon carbide had modified the microstructure of Al- Fe3O4 composite. As shown in Figure 3, proeutectic plates (Fe3O4) is more rectangular in shape, while in Figure 5, by adding silicon carbide, the microstructure has evolved from rectangular to a more spherical shape.
5. For Fig 5, the results of XRD do not match the above results.Thank you for this comment. Action has been done.
The XRD analysis in Figure 7 displays the phase identification in the specimens. The X-ray diffraction of two composites Al-15Fe3O4 analyses and Al-30Fe3O4-20SiC after sintering were shown. The measurements of X-ray diffraction after addition of reinforcements, were studied by comparing the peaks with diffraction. As can be seen, the diffraction peak with the highest intensity is related to Aluminium. Al-Fe intermetallic compound (Al3Fe, grey platelets) and Al2O3 were detected in Al-15Fe3O4 XRD analyses. In Figure 7, for Al-30Fe3O4-20SiC the first peak assigned to SiC (Hexagonal), and the second peak were referred to as Fe3O4 with a cubic crystallography system. Moreover, minor amount of Al-Fe intermetallic compounds (Al5Fe2 - Al3Fe), Fe3C cementite (iron carbide), Al2Fe3Si4 Iron Aluminium Silicon were identified. After adding hybrid reinforcement particles (Fe3O4-SiC), and heat treated, 2θ= (38/784, 44/600, 65/186, 78/306, 82/352) are related to aluminium, 2θ= (35/439, 43/070, 62/546) assigned to Fe3O4 cubic crystal system, and 2θ= (35/731, ,59/996, 71/944) related to SiC cubic crystal system. Moreover, Al2O3 has been detected.
Figure 7. Shows the X-ray diffraction (XRD) analyses for the two composites
6. In this paper, there are many selected components, but the results are not presented one by one.In this study, we tried to fabricate the components and design the experiments systematic so the lowest amount of component (Al-15Fe3O4), highest amount (Al-30Fe3O4-20SiC) and some composition between them are selected to find the trend of results so its help to select the optimum component. Moreover, some experiment adding lower amount of the hybrid reinforcement doesn’t have any significant effect on composite properties, based on that we disregard some tests for some samples. Although from each composition three samples had been prepared and the mean value of each experiment result has been represented.7. Please strengthen the logic of the description of the experimental results.
Thank you for this comment. Action has been done in mechanical, tribology and corrosion parts.
8. What is the effect of reinforced particles on the corrosion properties of materials?
Thank you for this helpful comment.
To explain more the better corrosion resistance of the hybrid composite at room temperature compared with the pure Al matix can explain with the fact that SiC particulates are ceramics and remain inactive in the artificial sea watersolution. They are barely influenced by the artificial sea water medium. According to previous researches the corrosion resistance of Aluminium matrix composite is controlled by many elements such as fabrication technique, characteristics of the matrix, the amount of the reinforcement, and the environmental features. Formation of Al4C3 at the reinforcement/matrix interface has negative effect on corrosion, while the undesirable Al4C3 has not been formed due to XRD analysis. Accordingly, the formation of Al4C3, Al2Fe3Si4 Iron Aluminium Silicon, at the SiC particles/matrix interface during fabrication is not likely to form. It is assumed that the SiC particles play a main role as a physical barrier. The particles play the role as a fairly barrier to the formation of corrosion pits [13]. The weakest part of composites is interface between the matrix and the reinforcement. Thereby, the interfacial bond, strong or weak, is important in the corrosion process [14].
Moreover, the observations revealed that Fe3O4 reinforced composites have higher corrosion resistance compare to aluminium matrix. Noteworthy, the detected pitting corrosion in aluminium base metal was not discovered in the existence of Fe3O4 nanoparticles. This might be in consequence of the existence of Al3Fe and Al5Fe2 intermetallics based on XRD and FESEM which act as cathodes with regard to metal matrix and enhance pitting corrosion resistance. The developed pitting behaviour is more essential in presence of chloride ions, while the aluminium surface becomes very vulnerable to pitting corrosion .Thus, it can be determined that Fe3O4 nanoparticles had a dual influence on the corrosion resistance of the aluminium matrix composite [15]. Based on the obtained results, it can be deduced that Al2O3 improves the corrosion resistance because Al2O3 affects the anodic and cathodic reactions [16].
9. Please describe in further detail the effect of enhanced particles on friction and wear properties and corrosion resistance.
A strong interfacial interaction between Al matrix and SiC with strong Al–Si bonding is reason of high wear resistance in the composite. In fact, Fe3O4-SiC particles with higher strength than Aluminium can withstand the applied load without much deformation. Thus, if the particles distribute finely, they can improve the wear properties. Moreover, Al3Fe intermetallic compounds reduce formability and fatigue resistance but improves wear resistance. The presence of the intermetallic compound with the high micro hardness will improve the micro hardness and wear resistance [17].

Reviewer 3 Report
This manuscript reports on the fabrication of composites consisting of aluminum matrix reinforced with Fe3O4-SiC hybrid nanofiller in different composition, and on the characterization of their properties form the point of view of microstructure, tribology and corrosion.
The subject of the manuscript is interesting and the wok seems to be systematic. However, there are some points that need to be improved. Regarding presentation, samples should be labelled always in the same way, or the same name should be given along the manuscript (see for instance figures 6 and 7). Otherwise at some points it is difficult to follow what is reported. It would also help to present the samples in the same order.
Additionally:
- In table 1, composition of samples 1 to 5 does not sum up 100% but 95%, while is 100% in the case of samples 6-8. Is that a mistake or does that mean that samples 6-8 do not contain 5% Mg stearate?
- Regarding the microstructural characterization, and the images of optical microscopy shown in figure 1 how one can be sure which areas correspond to a particular component without any further analysis? The same for FESEM images shown in figure 2.
- It would be appropriate to add error bars in figures 6 and 7
Author Response
Reviewer #3
1.Regarding presentation, samples should be labelled always in the same way, or the same name should be given along the manuscript (see for instance figures 6 and 7). Otherwise at some points it is difficult to follow what is reported. It would also help to present the samples in the same order.
Thank you for this comment. Action has been taken.
2.. In table 1, composition of samples 1 to 5 does not sum up 100% but 95%, while is 100% in the case of samples 6-8. Is that a mistake or does that mean that samples 6-8 do not contain 5% Mg stearate?
Thank you for this useful comment. The current presented data are accurate values. In the previous version of the manuscript a writing mistake has occurred with calculation of aluminum wt% which was modified as follows:
Table 1. Different compositions of aluminium, ferrous ferric oxide and silicon carbide
|
composition |
Al ( wt%) |
Fe3O4 ( wt%) |
SiC (wt%) |
|
sample 1 |
80 |
15 |
0 |
|
sample 2 |
70 |
15 |
10 |
|
sample 3 |
60 |
15 |
20 |
|
sample 4 |
50 |
15 |
30 |
|
sample 5 |
65 |
30 |
0 |
|
sample 6 |
55 |
30 |
10 |
|
sample 7 |
50 |
30 |
15 |
|
sample 8 |
45 |
30 |
20 |
3.. Regarding the microstructural characterization, and the images of optical microscopy shown in figure 1 how one can be sure which areas correspond to a particular component without any further analysis? The same for FESEM images shown in figure 2.
Thank you for this useful comment.
The optical microscopy images for Al-15Fe3O4 and Al-30Fe3O4-20SiC composite are presented in Figure 1a, and 1b respectively. Figure 1a shows homogenous distribution of Fe3O4 particles in Al matrix. Figure 1.b is SiC as grey colour element, and Fe3O4 particles as white colour element which is distributed quite uniformly in Al-30Fe3O4-20SiC sample.
Figure 1: Optical microscopy for two composition of (a) Al-15 Fe3O4 (b) Al-30 Fe3O4-20SiC
Efficient reinforcement requires well bonded principal with matrix and particles. Chemical bonding (covalent, metallic, ionic) inter-diffusion is the diffusion of atoms between two metals, and van der Waals bonding are the components of the interface mechanisms pertaining to the filler and matrix bonding, and the reaction among the matrix and reinforcements in composite.
The appropriate reaction among matrix and reinforcements assists wetting ability and bonding between them. The extreme reaction between particles and matrix may have an undesirable impact on the mechanical and thermal properties of the composite, while a severe reaction can damage the reinforcements [25].
Thereby, an ideal reaction is desired for composite fabrication. Magnetite is commonly found in self-sustaining thermite reaction. The Al–Fe3O4system is identified as highly exothermic reaction that can be employed during mechanical or thermal treatments based on the following stoichiometric reaction:
3Fe3O4 + 8Al → 4Al2O3 + 9Fe ΔH° = −3021 kJ (3)
One of the weak interface outcomes is reduction in stiffness, hardness and strength but highly resistant to fractures. In contrast, strong interface between particles in the matrix shows high stiffness and strength but typically low resistant to fracture. [26,27].Mechanical properties of the hybrid composite depends on the percentage of reinforcement materials, and microstructure and volume fraction of dendritic α-Al [28]. Adding silicon carbide has reformed the microstructure of Al-Fe3O4 composites, which improved mechanical properties.
In conformity with Ellingham–Richardson diagram, oxygen and aluminium reaction is more likely to occur compared to iron and aluminium, so Fe3O4 would be reduced by aluminium. Notwithstanding, due to the negative free energy of creation for various Al-Fe intermetallic between aluminium and iron there is a thermodynamic tendency to react with each other and form Al-Fe intermetallic compounds (IMCs) [29,30]. Based on binary phase diagram of Al-Fe ,two main phases of Al5Fe2 and Al3Fe were recognized at the interface of iron and aluminium, which have high hardness (Al3Fe = ~717 Hv, Al5Fe2 = ~944 Hv) [31]. Furthermore, there are various Al-Fe intermetallic compounds such as AlFe, Al2Fe, Al5Fe2, AlFe3 and Al3Fe based on Al-Fe phase diagram [30]. Although at temperature above 550°C, Al3Fe is the only stable phase since the sintering process occurred at 600°C.
The practical paths that Al–Fe3O4 reaction would proceed with are: (1) direct reaction of Fe3O4 to form Fe or (2) reduction of Fe3O4 through an intermediate reaction to form FeO, and then reduce to Fe. The reactions can be addressed as below [32]:
Fe3O4 → 3FeO + 1∕2O2 (4)
2Al + 3FeO → Al2O3 + 3Fe (5)
3Al + Fe → Al3Fe (6)
The phase diagram of Al-Fe-Si ternary system at 600◦C is shown in Figure 2 which illustrates that Al2Fe3Si4 is a stable phase at temperature above 600◦C.
Figure 2:Al-Fe-Si Computed Phase Diagram at 600◦C [33].
Figure 3 shows FESEM micrographs of Al-15Fe3O4 composite with uniform distribution Fe3O4 powders in different magnifications.
Figure 3: FE-SEM micrographs of Al-15Fe3O4 composite in different magnifications
Figure 4: The EDS spectrum of Al-15Fe3O4 composition (1) Aluminuim (2) Fe3O4 (3) Al3Fe(4) Al2O3
The EDS analysis results from four areas of Al-15Fe3O4 surface sample are shown in Figure 4 which were based on Figure 3. EDS analysis of the exposed surface point 1 shows the result for Aluminium with 85.63wt%. In point 2, Fe3O4 particle (white colour) exists in aluminium matrix with confirmed peaks of Fe (27.73wt%) and O (31.63wt%). Based on EDS analysis, it indicates that intermetallic phase of Al3Fe and interfaces of Al2O3occurred at selected points of 3 and 4 with Al 70.11wt%, Fe 27.73wt% and Al 85.65wt% were detected, respectively which were also confirmed in accordance to XRD analysis.
Figure 5: FESEM micrographs of Al-30Fe3O4-20SiC composite in different magnifications after etching
FESEM micrograph in different magnifications for Al-30Fe3O4-20SiC is also shown in Figure 5 composites after sintering at 600 °C and etched. Al matrix with homogeneous distribution of Fe3O4 powders (bright) and SiC particles (dark) in rectangular shape were positioned at the grain boundaries. Moreover, homogeneous distribution of particles in the matrix is evident. By increasing the amount of weight percentage of reinforcements, the possibility of agglomeration at grain boundaries is also increased. Figure 5c and d reveals a fully uniform distribution of particles, without any evidence of particle clustering.
The EDX analysis results from six areas of Al-30Fe3O4-20SiC sample surfaces which are shown in Figure 5 are then presented in Figure 6. The EDX results show Al, Fe, Si, O and Mg in the grain boundaries. Point 1 is Aluminium where Al peaks (82.83wt%) are the main peak and it is detected using FESEM in light grey colour. Point 2 is Fe3O4 (Fe 33.55 and O 42.35 wt%) in white particles and Point 3 is confirmed as SiC (Si 46.75, and C 10.7 wt%) in dark grey colour which were also confirmed by XRD pattern. According to EDS analyses, composition of Al3Fe (Al 57.76 and Fe 18.84 wt%), Al2O3 (Al 29.10 and O 45.15 wt%), and Al2Fe3Si4 (Al 20.5, Fe 30.76 and Si 41.25 wt%) at number 4, 5 and 6 were detected, respectivelywhich were also confirmed by XRD pattern. This analysis is also in agreement with previous researches [34-36].
Figure 6: The EDS spectrum of Al-30Fe3O4-20SiC composition(1) Aluminium(2) Fe3O4(3)SiC(4) Al3Fe(5) Al2O3(6) Al2Fe3Si4
By comparing microstructure images for Al-Fe3O4 and Al-Fe3O4-SiC composite in Figures 3 and 5, it is concluded that the addition of silicon carbide had modified the microstructure of Al- Fe3O4 composite. As shown in Figure 3, proeutectic plates (Fe3O4) is more rectangular in shape, while in Figure 5, by adding silicon carbide, the microstructure has evolved from rectangular to a more spherical shape.
4.. It would be appropriate to add error bars in figures 6 and 7
Thank you for this comment. Action has been taken.
We look forward to your positive response. If you have further question, please do not hesitate to contact me. We appreciate your kind consideration in this matter.

Round 2
Reviewer 1 Report
Accept.
Reviewer 3 Report
Manuscript has been modified according to the comments given in the previous revision